# VISOR: A Vision-Language Model-based Test Oracle for Testing Robots

### Prasun Saurabh
prasun@simula.no
Simula Research Laboratory and Oslo
Metropolitan University
Oslo, Norway

### Pablo Valle
pvalle@mondragon.edu
Mondragon University
Mondragon, Spain

### Aitor Arrieta
aarrieta@mondragon.edu
Mondragon University
Mondragon, Spain

### Shaukat Ali
shaukat@simula.no
Simula Research Laboratory
Oslo, Norway

### Paolo Arcaini
arcaini@nii.ac.jp
National Institute of Informatics
Tokyo, Japan

## Abstract

Testing robots requires assessing whether they perform their intended tasks correctly, dependably, and with high quality, a challenge known as the test oracle problem in software testing. Traditionally, this assessment relies on task-specific symbolic oracles for task correctness and on human manual evaluation of robot behavior, which is time-consuming, subjective, and error-prone. To address this, we propose VISOR, a Vision-Language Model (VLM)–based approach for automated test oracle assessment that eliminates the need of expensive human evaluations. VISOR performs automated evaluation of task correctness and quality, addressing the limitations of existing symbolic test oracles, which are task-specific and provide pass/fail judgments without explicitly quantifying task quality. Given the inherent uncertainty in VLMs, VISOR also explicitly quantifies its own uncertainty during test assessments. We evaluated VISOR using two VLMs, i.e., GPT and Gemini, across four robotic tasks on over 1,000 videos. Results show that Gemini achieves higher recall while GPT achieves higher precision. However, both models show low correlation between uncertainty and correctness, which prevents using uncertainty as a correctness predictor.

## CCS Concepts

• **Software and its engineering** → **Software testing and debugging**; • **Computing methodologies** → *Machine learning*.

## Keywords

Vision-Language Model, Robotics Manipulation, Test Oracles, Software Testing

**ACM Reference Format:**
Prasun Saurabh, Pablo Valle, Aitor Arrieta, Shaukat Ali, and Paolo Arcaini. 2026. VISOR: A Vision-Language Model-based Test Oracle for Testing Robots. In *Proceedings of 3rd ACM International Conference on AI-powered Software (AIware 2026)*. ACM, New York, NY, USA, 11 pages. https://doi.org/XXXXXXX.XXXXXXX

## 1 Introduction

Robots are increasingly being deployed across a wide range of applications in complex and dynamic environments, where they are expected to perform tasks correctly, meet performance and safety requirements, and avoid harming users and other entities [3, 26, 38, 42]. Moreover, robots are becoming increasingly autonomous [8] and are being deployed in safety and mission-critical applications, making their dependability and high-quality behavior extremely important.

This calls for systematic and automated testing of robots to ensure not only successful task completion but also adequate task quality, as defined by user criteria. In software testing, this problem is referred to as the *test oracle problem* [6], which determines whether software meets its intended behavior. In robotics, this means assessing whether a robot performs its intended task correctly and to an acceptable standard. Symbolic test oracles, which are commonly employed in this context [39, 50, 53], are typically task-specific, lack reusability across different tasks, and provide only binary pass/fail judgments, making them unable to automatically assess task quality. Task quality assessment often requires human involvement for validation [48], as humans visually inspect robot executions to judge task quality. This practice is time-consuming, error-prone, difficult to scale, and subjective, as different human testers may perceive task quality differently for the same test. As a result, testing robots at scale remains difficult without automation.

Vision-Language Models (VLMs) are large models capable of understanding and reasoning about images, videos, and text. They are increasingly being used for solving problems in several domains, including robotics [10, 11, 15, 46]. In this paper, we use VLMs to automate test oracle assessment for robots, by proposing a kind of VLM-as-a-judge approach called VISOR. The approach automatically assesses task correctness and quality from robot execution videos. Moreover, given the inherent uncertainty of VLMs, VISOR

explicitly quantifies this uncertainty, providing a measure of trustworthiness for each evaluation to determine whether the VLM's assessment can be relied upon or should be treated with caution.

We conducted a large-scale empirical analysis of VISOR using two VLMs, namely GPT and Gemini, on a dataset of over 1,000 videos spanning four different tasks of robotic arms, in which robots either successfully completed the tasks or failed to do so. We also define a new distance metric for the quality analysis that estimates the severity of misclassification in quality assessment. Experimental results reveal that Gemini excels at recall, while GPT produces more precise predictions aligned with ground truth. Both models exhibit low uncertainty and stable performance across different runs.

## 2 Background

In this section, we present the foundational concepts and background related to Vision-Language Models and Robotic Systems.

### 2.1 Vision-Language Models

VLMs represent a major advance in foundational multimodal artificial intelligence, enabling integrated perception, understanding, and reasoning across visual inputs (such as images or videos) and natural language [41, 49]. They are typically pre-trained on massive internet-scale datasets pairing images with descriptive captions, text, or other aligned multimodal pairs, fostering deep semantic alignments between visual content and linguistic representations [29]. Recent architectural advances have further enhanced their generality, efficiency, and scalability. Modern VLMs often integrate frozen high-capacity vision encoders (e.g., CLIP [41] and SigLIP [47]), powerful large language model backbones [22], and lightweight modality-fusion components such as projectors or adapters [29]. These advances support a wide variety of real-world applications [52], such as answering questions about medical images or documents [19] or content generation through summarization of infographics [4].

In robotics, VLMs serve as a powerful perception and high-level reasoning backbone, enabling robots to interpret their surroundings and understand natural language instructions. This has lead the emergence of Vision-Language-Action (VLA) models [8, 9], which extend VLMs by incorporating dedicated action-generation heads to control the robot. VLAs directly map combined visual and linguistic inputs to low-level robot control signals for embodied tasks [22, 23, 55].

### 2.2 Robotic Systems

Cyber-Physical Systems (CPSs) integrate computation, communication, and physical processes to sense, reason about, and act upon the real world [5, 12]. They operate in safety-critical domains such as autonomous vehicles [21], surgical and assistive medical robotics [13], and advanced manufacturing [35]. Robotic systems represent a core subclass of CPSs, where embodied agents must perceive dynamic environments, make decisions, and execute physical actions with precision and safety.

In this work, we focus on robotic manipulation systems, primarily articulated robotic arms fitted with end-effectors such as parallel-jaw grippers. They perform a wide variety of tasks, like grasping and placing objects. Reliable operation depends on tightly

integrated components: motor controllers for actuation, proprioceptive sensors for internal state estimation, exteroceptive sensors for environmental awareness, and planning-control algorithms that generate feasible, collision-free actions adapted to task goals. The recent irruption of VLA models [8, 9] significantly changed robotic manipulation systems. Traditional pipelines that handle perception, planning, and control separately can now be replaced by a single end-to-end learned model. A VLA-enabled robot can now directly interpret natural language instructions such as *"pick up the water bottle"*, while processing live visual observations to generate executable actions.

## 3 VISOR

Fig. 1 presents an overview of the proposed approach VISOR. We use a VLM as a test oracle to analyze videos of robots performing various tasks and assess their correctness and quality, rather than relying on manual human analysis of robots or of their videos for this purpose. In parallel, we explicitly quantify the VLM's uncertainty when performing these tasks, enabling interpretation of the reliability of automated task correctness and quality assessments.

In the context of this paper, such videos are produced by a simulator that takes as input initial scene data, such as the position (i.e., $x$, $y$, $z$) and rotation in quaternions (i.e., $q1$, $q2$, $q3$, $q4$) of each object in the scene. In addition to this, each object in the scene is represented with a $model_{id}$, which is used by the simulator to determine which is the object to be displayed since, for the same object (e.g., a can), different textures might be used (e.g., cola, fanta, 7up, . . . ). All this represents environmental configuration and together with the robot's initial pose, along with task instructions (e.g., "Pick up the orange"), constitute the inputs to the simulator. Next, the simulator, along with a control policy executes the task and generates a corresponding video, which is then assessed by a VLM to determine task correctness and quality. Such an assessment can determine whether a task has failed or succeeded. In the latter case, the VLM further assesses *task quality*, classifying it as *High*, *Medium*, or *Low*. Moreover, we measure the uncertainty of the task correctness prediction for each video by analyzing how confident it is in determining whether a task has been performed with success or fail. Similarly, for task quality assessment, we measure the uncertainty when distinguishing among *High*, *Medium*, and *Low* quality prediction. The specific metrics used to quantify uncertainty are described in Sect. 4.4.

A VLM requires a well-designed prompt to assess task correctness and quality of its execution [43]. To this end, we first made a small-scale evaluation of several prompting styles, including zero-shot and few-shot. Our preliminary experiments and analysis across these prompting styles revealed a substantial increase in the number of input tokens for few-shot prompting (i.e., 64K tokens) compared to zero-shot (i.e., 7K tokens), leading to significantly higher inference time and computational cost. As a result, few-shot prompting proved to be too expensive relative to the zero-shot approach. Therefore, we decided to use the zero-shot prompting style. We then iteratively refined the prompt to achieve more reliable outputs. Consequently, we designed two prompts: one for task correctness assessment (Fig. 2) and the other for test quality assessment (Fig. 3).

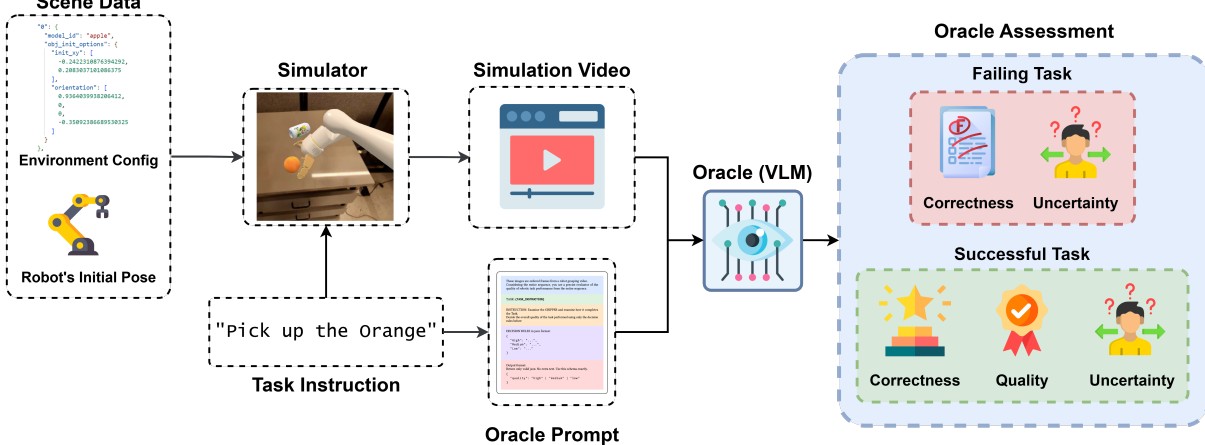

Figure 1: VISOR – In *Failing Task,* Correctness refers to failure; in *Successful Task,* Correctness refers to success.

These images are ordered frames from a robot grasping video.
You are a precise evaluator of quality of robotic task performance from entire sequence.

Decide if the **{TASK_INSTRUCTION}** performed by the robotic gripper arm was a Success or Failure in the given video.

Output format: Return only valid JSON. No extra text. Use this schema exactly.

```
{
    "status": "Successful" | "Failure"
}
```

Figure 2: Prompt template for task correctness assessment.

These images are ordered frames from a robot grasping video.
Considering the entire sequence, you are a precise evaluator of the quality of robotic task performance from the entire sequence.

TASK: **{TASK_INSTRUCTION}**

INSTRUCTION: Examine the GRIPPER and examine how it completes the Task.
Decide the overall quality of the task performed using only the decision rules below:

DECISION RULES in JSON format:

```
{
    "High": "...",
    "Medium": "...",
    "Low": "..."
}
```

Output format:
Return only valid json. No extra text. Use this schema exactly.

```
{
    "quality": "high" | "medium" | "low"
}
```

Figure 3: Prompt template for task quality assessment.

Fig. 2 shows the prompt template for task correctness assessment. The first part of the prompt assigns the VLM the role of a test oracle and provides context by specifying that video frames are presented as an ordered sequence. It also includes task instructions for each video, e.g., "Pick up the Orange" specified in the *TASK_INSTRUCTION* variable, along with details on what the VLM must observe and classify. Finally, the prompt enforces output requirements, requiring the VLM to generate a JSON-only schema with a binary label (i.e., Successful or Failure).

Fig. 3 shows the prompt template we used for task quality assessment. Like the task correctness prompt, it assigns the VLM the role of a test oracle and provides contextual information. Next, it provides a task instruction (e.g., "Pick up the Orange"), specified in the *TASK_INSTRUCTION* variable, along with details on the task quality to be assessed. It also defines predefined decision rules, which are the ones Valle et. al. [48] used to guide the human assessment in their experiment, that map task execution to three quality levels (i.e., *High*, *Medium*, or *Low*). Finally, it requires the VLM to produce a strict JSON-only output using a fixed schema.

## 4 Empirical Evaluation

We conducted an empirical study to assess the effectiveness of VLMs in evaluating the correctness and quality of robotic tasks. This section presents our research questions and details of the

experimental setup used for the evaluation. All scripts, benchmarks, and results are available at [44].

### 4.1 Research Questions (RQs)

Our overall objective is to evaluate the effectiveness of VLMs as automated test oracles for robotic systems by assessing their ability to determine task success and task quality. Based on this objective, we aim to address the following research questions (RQs):

**RQ1** *How effective are VLMs as automated test oracles for robotic task execution?*

This research question investigates whether VLMs can effectively serve as test oracles for assessing task success or failure (RQ1.a) and task quality (RQ1.b). To this end, we further subdivide this RQ into two sub-questions.

**RQ1.a** *How accurately can VLMs determine task success or failure from robot execution videos for different tasks?*

This RQ evaluates the performance of VLMs as classifiers

for the task correctness by labeling robot execution videos as either successful or failed.

**RQ1.b** *How effectively can VLMs assess the quality of successfully completed robotic tasks?*

This RQ evaluates the performance of VLMs in classifying the quality of successful executions using three categories: *High*, *Medium*, or *Low*. This results in a multi-class classification problem to be solved by VLMs.

**RQ2** *How uncertain are VLMs in their assessments of task correctness and task quality?*

This RQ investigates the uncertainty of VLM predictions, that is, how confident the VLMs are for both task correctness and task quality evaluations.

**RQ3** *To what extent does VLM uncertainty correlate with test oracle decisions for task correctness and quality assessment?*

Studying this relationship is important to determine whether uncertainty can serve as a reliable indicator of assessments of task correctness and task quality. It also helps identify which uncertainty metrics are most suitable for evaluating the trustworthiness of VLMs for test oracle assessment.

## 4.2 Evaluation Dataset

For our evaluation, as a ground truth dataset, we used the videos provided by Valle et al. [48], which comprise four tasks (i.e., MoveNear, PickUp, PutIn, and PutOn) across three Visual Language Action (VLA) models (i.e., OpenVLA [23], $\pi_0$ [9], and SpatialVLA [40]). The evaluation has 500 videos per task and VLA model, forming an unbalanced dataset of successful and failing task executions. Moreover, among successful executions, the distribution of task quality (i.e., *High*, *Medium*, *Low*) is also uneven. To address this, we curated a balanced subset for our evaluation as follows.

For each task, we first determined the maximum number of successful videos available at each task quality, across all VLAs. For example, for PickUp, the numbers of successful videos are 201 (*High*), 66 (*Medium*), and 117 (*Low*); so, for this task, we set the maximum number of videos for each quality level as 66. We then selected 66 videos for each quality, attempting to balance across VLAs. Ideally, this would be 22 videos per VLA model, per quality level. However, for models with fewer than 22 videos, we included all their videos and distributed the remaining videos among the other models. For instance, for *High* quality videos for PickUp, we selected the 18 videos from OpenVLA, and we balanced the remaining ones by selecting 24 each from $\pi_0$ and SpatialVLA.

Applying this procedure across all tasks, we obtained 198 videos for PickUp, 141 for MoveNear, 66 for PutIn, and 111 for PutOn, totaling 516 successful videos. To balance the dataset, we included an equal number of failing videos per task, that is 198, 141, 66, and 111, respectively. Since sufficient failing videos were available, this selection could be fully balanced across models. The resulting evaluation dataset comprised 1,032 videos.

## 4.3 Selected Vision-Language Models

We conducted a small exploratory experiment to select the VLMs for our study. We assessed six VLMs from open-source and proprietary categories: GPT-4.1 [37], Gemini-2.5-Flash [17], LLaVA-Next [25], SmolVLM2 [34], Qwen [51], and Eagle [45]. For a fair comparison,

we used the same settings across all VLMs, including a temperature of 0 to minimize output stochasticity. We ran the experiment five times, selecting six video files from each of the four tasks.

For task correctness assessment GPT-4.1 and Gemini-2.5-Flash produced more consistent and task-sensitive results across the four tasks. In contrast, the open-source models (LLaVA-Next, SmolVLM2, Qwen, and Eagle) exhibited identical scores (i.e., Precision=0.500, Recall=1.000, F1=0.667) across all tasks, indicating a behavior that does not reliably identify the outcome. We observed that all open-source VLMs predicted all videos as "Successful" in the correctness assessment, never resulting in a "False Negative". Consequently, the higher recall results from this consistent prediction behavior rather than evidence of strong task understanding, and it does not do justice to the skewed outcome. Because on this lack of task sensitivity, we excluded these open-source models from the full experiments, and we only used GPT-4.1 and Gemini-2.5-Flash (called GPT and Gemini in the following).

## 4.4 Evaluation Metrics

**Evaluation Metrics for RQ1.a.** We apply usual binary classification metrics, i.e., *precision*, *recall*, and *F1*, to assess the performance of the selected VLMs per task. Since each experiment is repeated 10 times, we report the mean and standard deviation of each metric to show variability in effectiveness across runs. To assess the statistical significance, we apply the Mann-Whitney U test for each metric and task. A *p*-value less than 0.05 indicates statistically significant differences. In addition, we report the Vargha-Delaney effect size $\hat{A}_{12}$ to quantify the magnitude of the observed differences.

For each metric (i.e., precision, recall, or F1), $\hat{A}_{12}$ estimates the probability that one VLM (V1) yields higher values than another VLM (V2). $\hat{A}_{12} = 0.5$ indicates no difference, while values greater than 0.5 indicate that V1 is more likely to achieve higher metric values than V2. We interpret the results following the literature [33]: 1) negligible $\hat{A}_{12} \in (0.44, 0.56)$, 2) small for $\hat{A}_{12} \in (0.34, 0.44]$ or $\hat{A}_{12} \in [0.56, 0.64)$, 3) medium for $\hat{A}_{12} \in (0.29, 0.34]$ or $\hat{A}_{12} \in [0.64, 0.71)$, and 4) large for $\hat{A}_{12} \in [0, 0.29]$ or $\hat{A}_{12} \in [0.71, 1]$.

**Evaluation Metrics for RQ1.b.** We evaluate the VLMs performance on the multi-class task of assessing task quality with multi-class classification metrics, i.e., Precision-micro, Precision-macro, Recall-micro, Recall-macro, F1-micro, and F1-macro. The macro versions of these metrics treat each class equally, providing an overall measure of effectiveness across all quality categories (e.g., *High*, *Medium*, *Low*), while the micro versions aggregate contributions from all samples, giving greater influence to classes with more instances. Using both versions of the metrics allows us to evaluate the overall performance of the VLMs across classes and their performance under class imbalance.

In addition, we define a *distance* metric $d \in \{0, \dots, 5\}$ to quantify the discrepancy between a predicted task outcome and the ground truth by considering both task correctness and task quality. Let $c \in \{pass, fail\}$ denote the ground-truth correctness and $\hat{c}$ the predicted correctness. The correctness component of the distance is defined as $d_c = 0$ if $\hat{c} = c$, and $d_c = 2$ otherwise. Task quality is assumed to be ordinal with levels $q \in \{Failure, Low, Medium, High\}$, which we map to numerical values $\{0, 1, 2, 3\}$. Let $\hat{q}$ denote the predicted quality level; the quality component of the distance is then $d_q = |\hat{q} - q|$. The

final distance is computed as the sum $d = d_c + d_q$, which ranges from 0 for a perfect prediction to 5 for a maximally incorrect prediction.

To assess statistical significance between the two selected VLMs for each metric, we apply the Mann–Whitney U test and report the Vargha–Delaney $\hat{A}_{12}$ effect size, as in RQ1.a. For the distance metric, we compute, for each task, run and VLM, the mean distance across all videos, where lower values indicate better performance. For all other metrics, higher values correspond to better performance.

**Evaluation Metrics for RQ2.** We analyze the uncertainty of the models in order to determine how reliable their answer is. We selected one uncertainty metric derived from model outputs, i.e., Entropy [7], which quantifies the dispersion of the predicted probability distribution over output classes. We also compute commonly used confidence-based metrics, i.e., Maximum Softmax Probability (MSP) and DeepGini [16], defined as the difference between the highest and second-highest predicted class probabilities. Larger margins indicate more confident predictions, whereas smaller margins suggest higher uncertainty.

**Evaluation Metrics for RQ3.** In RQ3, we focus on finding the relationship between uncertainty and performance of the models. Specifically, we analyze the correlation between the proposed distance metric and the uncertainty measures from RQ2. By focusing on distance rather than accuracy alone, we account for the severity of prediction errors and examine whether higher uncertainty is associated with larger deviations from the ground truth. We perform Spearman's correlation test to quantify the strength and direction of this relationship, thereby assessing whether uncertainty estimates can serve as reliable indicators of prediction quality. The value of the correlation coefficient ($\rho$) ranges from -1 to +1, where negative values indicate a negative correlation and positive values indicate a positive correlation. We also report the $p$-value, where a value lower than 0.05 indicates a statistically significant correlation.

## 5 Analysis of the Results

In this section, we present a detailed analysis of the results obtained for each of the research questions posed in this paper. Our goal is to evaluate the capabilities of VLMs as test oracles, highlighting both their strengths and limitations in assessing robotic task execution.

### 5.1 RQ1: Effectiveness of VLMs as Test Oracles

This research question assesses the effectiveness of VLMs as test oracles. Specifically, we aim to determine how well these models evaluate two key aspects of task performance: (i) *correctness*, i.e., whether the task was completed as intended, and (ii) *quality*, i.e., how well the task was executed according to defined standards.

*5.1.1 RQ1a – Effectiveness of VLMs as Test Oracle for Judging Task Correctness.* As shown in Table 1, for MoveNear, Gemini achieves a precision of 0.561 and a high recall of 0.730, resulting in an overall F1 score of 0.635. This indicates that Gemini correctly classifies most test assessments but produces some false positives. In contrast, GPT exhibits high precision (0.844) but low recall (0.251), yielding a low F1 score of 0.386. While GPT makes very few false positives, it misses many correct test assessments. As shown in Table 2, statistical tests confirm that GPT significantly outperforms Gemini in precision with a large effect size, whereas Gemini is significantly better than GPT

**Table 1: RQ1.a – Precision, recall, and F1 scores of** GPT **and** Gemini **for the four tasks for task correctness.**

| VLM | Task | Precision | | Recall | | F1 | |
|---|---|---|---|---|---|---|---|
| | | $m$ | $\sigma$ | $m$ | $\sigma$ | $m$ | $\sigma$ |
| GPT | MoveNear | 0.844 | 0.044 | 0.251 | 0.016 | 0.386 | 0.019 |
| | PickUp | 0.775 | 0.007 | 0.541 | 0.018 | 0.637 | 0.013 |
| | PutIn | 0.980 | 0.000 | 0.739 | 0.027 | 0.842 | 0.018 |
| | PutOn | 0.959 | 0.015 | 0.746 | 0.018 | 0.839 | 0.013 |
| Gemini | MoveNear | 0.561 | 0.014 | 0.730 | 0.000 | 0.635 | 0.009 |
| | PickUp | 0.609 | 0.007 | 0.836 | 0.003 | 0.704 | 0.004 |
| | PutIn | 0.782 | 0.053 | 0.651 | 0.031 | 0.711 | 0.041 |
| | PutOn | 0.674 | 0.030 | 0.869 | 0.005 | 0.759 | 0.021 |

**Table 2: RQ1.a – Mann-Whitney U-Test and $\hat{A}_{12}$ statistics to compare** GPT **vs** Gemini **in terms of task task correctness.**

| Task | Precision | | Recall | | F1 | |
|---|---|---|---|---|---|---|
| | $p$-value | $\hat{A}_{12}$ | $p$-value | $\hat{A}_{12}$ | $p$-value | $\hat{A}_{12}$ |
| MoveNear | 0.000144 | 1.00 | < 0.0001 | 0.00 | 0.000144 | 0.00 |
| PickUp | 0.000144 | 1.00 | 0.000142 | 0.00 | 0.000144 | 0.00 |
| PutIn | 0.000141 | 1.00 | 0.000142 | 1.00 | 0.000142 | 1.00 |
| PutOn | 0.000145 | 1.00 | 0.000141 | 0.00 | 0.000145 | 1.00 |

for recall and F1, also with large effect sizes. Generally, the standard deviations are very low (Table 1), indicating that the performance of the models across different runs is stable.

As shown in Table 1, for PickUp, Gemini obtained a precision of 0.609 and a high recall of 0.836, resulting in an F1 score of 0.704. This suggests that Gemini captures most of the correct test assessments, though it also produces a moderate number of false positives. In comparison, GPT demonstrates higher precision (i.e, 0.775) but lower recall (i.e., 0.541) than Gemini, yielding a lower F1 score of 0.637. Furthermore, as Table 2 depicts, the results are the same as for MoveNear, i.e., GPT significantly outperforms Gemini in precision with a large effect size, whereas Gemini is significantly better in recall and F1 than GPT, also with large effect sizes. The low standard deviations (Table 1) indicate that overall both models' performance is consistent and stable across different runs.

Table 1 shows that for PutIn, GPT outperforms Gemini in precision (0.980 vs. 0.782), recall (0.739 vs. 0.651), and F1 score (0.842 vs. 0.711). As reported in Table 2, these differences are statistically significant, with large effect sizes across all three metrics. For PutOn, GPT achieves higher precision (0.959 vs. 0.674) and F1 score (0.839 vs. 0.759) than Gemini, but lower recall (0.746 vs. 0.869). Statistical tests in Table 2 confirm these differences, with all comparisons statistically significant and large effect sizes. All standard deviation values are low (Table 1), indicating stable performance across runs.

Based on these results, we conclude that across the four tasks, Gemini typically achieves higher recall, capturing more true test assessments, while GPT generally exhibits higher precision, producing fewer false positives. Thus, we suggest that using both models

**Table 3: RQ1.b – Precision/recall/F1 (micro and macro) and distance of** `GPT` **and** `Gemini` **for the four tasks (multiclass task-quality).**

| VLM | Task | Precision-micro | | Precision-macro | | Recall-micro | | Recall-macro | | F1-micro | | F1-macro | | Distance | |
|---|---|---|---|---|---|---|---|---|---|---|---|---|---|---|---|
| | | $m$ | $\sigma$ | $m$ | $\sigma$ | $m$ | $\sigma$ | $m$ | $\sigma$ | $m$ | $\sigma$ | $m$ | $\sigma$ | $m$ | $\sigma$ |
| GPT | MoveNear | 0.193 | 0.0232 | 0.168 | 0.0365 | 0.193 | 0.0232 | 0.220 | 0.0208 | 0.193 | 0.0232 | 0.164 | 0.0264 | 1.856 | 0.0482 |
| | PickUp | 0.390 | 0.0175 | 0.267 | 0.0273 | 0.390 | 0.0175 | 0.344 | 0.0175 | 0.390 | 0.0175 | 0.265 | 0.0191 | 1.534 | 0.0389 |
| | PutIn | 0.386 | 0.0219 | 0.266 | 0.0218 | 0.386 | 0.0219 | 0.301 | 0.0163 | 0.386 | 0.0219 | 0.270 | 0.0172 | 0.847 | 0.0475 |
| | PutOn | 0.483 | 0.0214 | 0.340 | 0.0379 | 0.483 | 0.0214 | 0.383 | 0.0179 | 0.483 | 0.0214 | 0.333 | 0.0231 | 0.843 | 0.0345 |
| Gemini | MoveNear | 0.205 | 0.0015 | 0.152 | 0.0032 | 0.205 | 0.0015 | 0.262 | 0.0021 | 0.205 | 0.0015 | 0.155 | 0.0022 | 2.506 | 0.0015 |
| | PickUp | 0.222 | 0.0000 | 0.141 | 0.0000 | 0.222 | 0.0000 | 0.270 | 0.0000 | 0.222 | 0.0000 | 0.150 | 0.0000 | 2.418 | 0.0000 |
| | PutIn | 0.301 | 0.0000 | 0.221 | 0.0000 | 0.301 | 0.0000 | 0.297 | 0.0000 | 0.301 | 0.0000 | 0.182 | 0.0000 | 1.794 | 0.0000 |
| | PutOn | 0.276 | 0.0000 | 0.217 | 0.0000 | 0.276 | 0.0000 | 0.322 | 0.0000 | 0.276 | 0.0000 | 0.195 | 0.0000 | 2.153 | 0.0000 |

**Table 4: RQ1.b – Mann-Whitney U-Test and $\hat{A}_{12}$ statistics to compare** `GPT` **vs** `Gemini` **in terms of task quality.**

| Task | Precision-micro | | Precision-macro | | Recall-micro | | Recall-macro | | F1-micro | | F1-macro | | Distance | |
|---|---|---|---|---|---|---|---|---|---|---|---|---|---|---|
| | $p$-value | $\hat{A}_{12}$ | $p$-value | $\hat{A}_{12}$ | $p$-value | $\hat{A}_{12}$ | $p$-value | $\hat{A}_{12}$ | $p$-value | $\hat{A}_{12}$ | $p$-value | $\hat{A}_{12}$ | $p$-value | $\hat{A}_{12}$ |
| MoveNear | 0.0189 | 0.20 | 0.1426 | 0.69 | 0.0189 | 0.20 | < 0.0001 | 0.00 | 0.0189 | 0.20 | 0.0433 | 0.76 | < 0.0001 | 0.00 |
| PickUp | < 0.0001 | 1.00 | < 0.0001 | 1.00 | < 0.0001 | 1.00 | < 0.0001 | 1.00 | < 0.0001 | 1.00 | < 0.0001 | 1.00 | < 0.0001 | 0.00 |
| PutIn | < 0.0001 | 1.00 | < 0.0001 | 1.00 | < 0.0001 | 1.00 | 1.0000 | 0.50 | < 0.0001 | 1.00 | < 0.0001 | 1.00 | < 0.0001 | 0.00 |
| PutOn | < 0.0001 | 1.00 | < 0.0001 | 1.00 | < 0.0001 | 1.00 | < 0.0001 | 1.00 | < 0.0001 | 1.00 | < 0.0001 | 1.00 | < 0.0001 | 0.00 |

simultaneously may be beneficial to improve test assessment performance. Their outputs, for example, through a voting mechanism, could then be used to determine the final test assessment.

*5.1.2 RQ1.b – Effectiveness of VLMs as Test Oracle for Judging Task Quality.* Table 3 summarizes the results of assessing VLMs for judgment test quality. Overall, precision, recall, and F1 (both micro and macro) are relatively low across tasks. For example, the best performance is observed for the `PutOn` task using GPT, with a precision-micro of 0.483, recall-micro of 0.483, and F1-micro of 0.483. For the remaining tasks, and across both models, the other metric values are even lower, indicating that the exact multiclass judgment test assessment remains challenging for the evaluated VLMs. The main reason behind this challenge is that the current VLMs are limited in reasoning about spatial changes in the datasets. Moreover, the three quality classes *High*, *Medium*, and *Low* often differ by only fine-grained execution. This is even more evident in the `MoveNear` task, where detecting the spatial gap proved more difficult.

When comparing the two models (Table 4), for `PickUp`, `PutIn`, and `PutOn`, GPT outperforms `Gemini` on all metrics with large effect sizes, except for recall-macro on `PutIn` where there are no significant differences. In contrast, for the `MoveNear` task, `Gemini` significantly outperformed GPT on precision-micro, recall-micro, recall-macro, and F1-micro, with large effect sizes. There were no significant differences between the models for precision-macro; however, GPT achieved significantly higher F1-macro than `Gemini`.

However, when examining the distance metric, we can see that for `Gemini`, the lowest (best) distance is achieved on `PutIn` (1.794), while the highest (worst) value is observed for `MoveNear` (2.506). For GPT, the best performance is again obtained on `PutOn`, with a substantially lower distance of 0.843, whereas the worst performance occurs on `MoveNear`, with a distance of 1.856. This suggests

that although the models often fail to predict the exact quality class, their predictions are generally close to the true quality levels. GPT exhibits significantly lower distances than `Gemini`, with large effect sizes, as shown in Table 4, suggesting that GPT produces judgment test quality assessments that are closer to the ground truth.

**Answer to RQ1:** Our results suggest complementary strengths between the two VLMs: `Gemini` excels at identifying the most relevant test assessments (high recall), whereas GPT produces more precise predictions closer to the ground truth.

*5.1.3 Qualitative analysis of misclassification.* We conducted a qualitative analysis to understand `VISOR`'s limitations in classifying task correctness and task quality.

*Task Correctness misclassification:* We observed three recurrent patterns in task-correctness misclassification:

(1) **Near-complete executions were classified as success,** and borderline cases, where the robotic arm almost placed the object onto the target but did not fully complete the task, were often still judged as successful.

(2) **The wrong object interaction was classified as success.** In some cases, the robotic arm interacted with the wrong object, but the VLM still predicted success. For example, the arm picked up the Coca-Cola can and placed it in the basket, although the task required placing the Pepsi can.

(3) **Task-like motion classified as success.** The robotic arm sometimes appeared to perform the intended motion without fully completing one part of the task, yet such executions were still classified as successful.

These patterns were especially pronounced for `Gemini`, which misclassified failure videos as success more frequently than GPT, particularly for *MoveNear* and *PickUp*.

**Table 5: RQ2 – DeepGini, Entropy, and MSP of GPT and Gemini for the four tasks for task correctness. ↑ means a high value represents low uncertainty and vice versa.**

| VLM | Task | DeepGini ↓ | | Entropy ↓ | | MSP ↑ | |
|---|---|---|---|---|---|---|---|
| | | m | σ | m | σ | m | σ |
| GPT | MoveNear | 0.0733 | 0.0089 | 0.0311 | 0.0041 | 0.9789 | 0.0027 |
| | PickUp | 0.0445 | 0.0050 | 0.0192 | 0.0023 | 0.9865 | 0.0017 |
| | PutIn | 0.0222 | 0.0064 | 0.0090 | 0.0029 | 0.9942 | 0.0021 |
| | PutOn | 0.0447 | 0.0071 | 0.0193 | 0.0036 | 0.9865 | 0.0031 |
| Gemini | MoveNear | 0.0007 | 0.0001 | 0.0001 | 0.0000 | 0.9999 | 0.0000 |
| | PickUp | 0.0003 | 0.0001 | 0.0001 | 0.0000 | 1.0000 | 0.0000 |
| | PutIn | 0.0006 | 0.0001 | 0.0001 | 0.0000 | 1.0000 | 0.0000 |
| | PutOn | 0.0004 | 0.0001 | 0.0001 | 0.0000 | 1.0000 | 0.0000 |

**Table 6: RQ2 – DeepGini, Entropy, and MSP of GPT and Gemini for the four tasks for task-quality assessment. ↑ means a high value represents low uncertainty and vice versa.**

| VLM | Task | DeepGini ↓ | | Entropy ↓ | | MSP ↑ | |
|---|---|---|---|---|---|---|---|
| | | m | σ | m | σ | m | σ |
| GPT | MoveNear | 0.0978 | 0.0369 | 0.1599 | 0.0568 | 0.9281 | 0.0303 |
| | PickUp | 0.0322 | 0.0174 | 0.0536 | 0.0274 | 0.9770 | 0.0135 |
| | PutIn | 0.0468 | 0.0315 | 0.0812 | 0.0502 | 0.9683 | 0.0242 |
| | PutOn | 0.0585 | 0.0176 | 0.0974 | 0.0246 | 0.9592 | 0.0152 |
| Gemini | MoveNear | 0.0008 | 0.0005 | 0.0026 | 0.0014 | 0.9996 | 0.0003 |
| | PickUp | 0.0013 | 0.0015 | 0.0032 | 0.0026 | 0.9992 | 0.0009 |
| | PutIn | 0.0012 | 0.0011 | 0.0033 | 0.0023 | 0.9994 | 0.0006 |
| | PutOn | 0.0002 | 0.0001 | 0.0010 | 0.0004 | 0.9999 | 0.0000 |

*Task Quality misclassification:* For task-quality prediction, we observed three recurrent failure modes:

(1) **Overlooking subtle execution defects:** cues such as hesitation, vibration, unstable grasp, slight collision, or brief loss of control were often missed by the VLM, thus affecting the task quality classification.

(2) **Ignoring mid-execution quality-relevant events:** the VLM sometimes emphasized the final state of the task while overlooking events occurring in the middle of the execution, which affected the overall quality.

(3) **Borderline and late-stage errors:** the VLM struggled with borderline cases and with issues arising near the end of the execution, such as dropping or hitting the object, often predicting a higher quality class than intended.

## 5.2 RQ2: Uncertainty Assessment

Table 5 presents descriptive statistics for Gemini and GPT across the four tasks for the three uncertainty metrics. The results show that for DeepGini and Entropy, the values are very close to zero, whereas for MSP, all values are near 1.0, indicating that both models exhibit extremely low uncertainty for all the tasks. Moreover, the very low standard deviations across all metrics indicate that both models are highly stable across runs. Similar patterns are observed for task quality assessment across all four tasks (Table 6), where uncertainty remains low, and model uncertainty is consistently stable across runs (low standard deviation).

**Table 7: RQ3 – Spearman correlation between distance and uncertainty metrics (DeepGini, Entropy, and MSP). Higher DeepGini and Entropy values indicate higher uncertainty, whereas higher MSP values indicate lower uncertainty.**

| VLM | Task | DeepGini | | Entropy | | MSP | |
|---|---|---|---|---|---|---|---|
| | | p-value | ρ | p-value | ρ | p-value | ρ |
| GPT | MoveNear | 0.8105 | 0.0088 | 0.8699 | 0.0060 | 0.7884 | −0.0099 |
| | PickUp | < 0.0001 | 0.2193 | < 0.0001 | 0.2193 | < 0.0001 | −0.2194 |
| | PutIn | 0.6558 | −0.0187 | 0.6569 | −0.0186 | 0.6573 | 0.0186 |
| | PutOn | < 0.0001 | 0.1657 | < 0.0001 | 0.1666 | < 0.0001 | −0.1657 |
| Gemini | MoveNear | 0.0003 | 0.0776 | 0.0003 | 0.0784 | 0.0003 | −0.0776 |
| | PickUp | 0.0077 | 0.0482 | 0.0055 | 0.0502 | 0.0077 | −0.0482 |
| | PutIn | < 0.0001 | 0.3120 | < 0.0001 | 0.3120 | < 0.0001 | −0.3120 |
| | PutOn | 0.0297 | 0.0539 | 0.0192 | 0.0580 | 0.0297 | −0.0539 |

The consistently low uncertainty values observed across all tasks and both models should be interpreted with caution. While low uncertainty may indicate confident predictions, it does not necessarily imply correctness, particularly for task quality assessment, where effectiveness metrics remain modest. This suggests that the VLMs are often highly confident even when their predictions deviate from the ground truth, pointing to potential overconfidence in their internal probability estimates. This behavior is especially evident when contrasting the low uncertainty values in Tables 5 and 6 with the relatively high distance values reported for task quality assessment. From a test oracle perspective, this highlights an important limitation: uncertainty estimates alone are insufficient to guarantee correctness and should not be interpreted in isolation. Nevertheless, the high stability of uncertainty metrics across runs indicates that uncertainty estimation itself is reproducible and consistent, making it a viable signal to be combined with effectiveness metrics or post-hoc calibration techniques when assessing the reliability of VLM-based test oracles for robotic manipulation applications.

> **Answer to RQ2:** Both models exhibit consistently low uncertainty and stable performance across tasks. However, low uncertainty does not imply correctness, and models may exhibit high confidence even when classifying incorrectly. Thus, uncertainty metrics should be considered with effectiveness metrics for reliability and correctness.

## 5.3 RQ3: Correlation Correctness-Uncertainty

Table 7 reports Spearman's rank correlations between the uncertainty metrics and distance. For Gemini, DeepGini, and Entropy exhibit significant positive correlations ($p < 0.05$, $\rho > 0$), indicating that higher uncertainty measured by these metrics is associated with larger distances, while lower uncertainty corresponds to smaller distances. For MSP, the same pattern is observed across all four tasks; however, since MSP is interpreted inversely, the correlations are negative ($\rho < 0$). Overall, these results suggest that higher uncertainty in Gemini is generally associated with predictions that deviate more from the ground truth.

For GPT, similar patterns to Gemini are observed for MoveNear, PickUp, and PutOn across the three uncertainty metrics. However, for MoveNear, the correlations are not statistically significant ($p > 0.05$). In contrast, for PutIn, the direction of the correlation is reversed, indicating a negative association between uncertainty

and distance; however, the correlation coefficient is very close to zero and not statistically significant ($p > 0.05$).

The correlation analysis reveals that uncertainty can serve as a meaningful, although imperfect, indicator of test oracle reliability. For Gemini, the consistent and statistically significant correlations across tasks suggest that higher uncertainty is generally aligned with poorer oracle decisions, as reflected by larger distances from the ground truth. This indicates that, for this model, uncertainty estimates provide useful information about when its assessments should be treated with caution. In contrast, GPT exhibits weaker and less consistent correlation, with some tasks showing no statistically significant relationship between uncertainty and distance. This discrepancy suggests that the usefulness of uncertainty as a proxy for correctness might be model-dependent and influenced by how uncertainty is internally represented and exposed. Notably, the absence of significant correlations for certain tasks implies that low uncertainty does not always guarantee accurate predictions.

> **Answer to RQ3:** Uncertainty and distance are slightly correlated, with Gemini showing consistent positive correlations and GPT exhibiting similar but even weaker trends.

## 6 Threats to Validity

Threats to conclusion validity are mainly related to the stochastic nature of the models used, even when the temperature is set to 0. To mitigate these effects, we ran each task 10 times to account for variability and applied appropriate statistical tests in accordance with established guidelines in the literature [2].

Threats to internal validity arise from the models' parameter settings. We configured them identically to enable a fair comparison. For example, we set the temperature parameter to 0 for both models. Moreover, we used the same prompt template for both models.

Threats to construct validity arise from the evaluation metrics used. For task correctness and task quality assessment, we used well-established metrics from the machine learning domain. Moreover, to capture the extent to which predictive quality deviates from the ground truth, we defined a distance-based metric.

A threat to external validity could be that we use four tasks, around 1,000 videos, and two models. However, this is a sufficiently large number of videos extracted from three state-of-the-art vision language action model-based robots. Including a broader set of tasks and videos would strengthen our findings across diverse tasks, while evaluating additional VLM models would help assess how well the results generalize beyond those considered in this study.

Another external validity threat is that VISOR has been tested only on simulation datasets. While VISOR can be adapted to real-world use cases, how well it performs is an area of future investigation. It is expected that real-world settings will introduce additional challenges like sensor noise, motion blur, etc. These factors may further affect the results of task-correctness and task-quality. Therefore, our current claims are limited to simulation-recorded datasets.

## 7 Discussion

This paper assessed the feasibility of using VLMs as test oracles for robotic tasks testing by analyzing task execution videos generated from simulation. Although we were unable to use open-source

VLMs effectively, proprietary VLMs performed well in assessing task correctness and quality, with low uncertainty. Moreover, the results were consistent across multiple runs, indicating reliable performance. However, we observed that model uncertainty remained low even when the VLM failed at properly determining task correctness and quality, suggesting that the employed uncertainty metrics may not reliably reflect prediction confidence. As a result, we foresee the need to define new uncertainty metrics that can be reliably used to assess task correctness and quality.

Moreover, we see potential to reduce costs using VLMs as a test oracle for robotic task testing. In our experiments, the per-test cost associated with monetary costs is approximately $0.03 per assessment and inference time of approximately 7 seconds. Compared to manual assessment, where a human evaluates a task in real-time, and for the failing cases there is no need to finish the task assessment, VISOR must wait for the simulation to complete, resulting in inference time and monetary cost. For a single video or a small set of videos, human assessment is generally faster and more reliable. However, as the number of videos to evaluate increases, the advantage of using VISOR becomes more significant. Apart from being able to evaluate videos continuously for days, in environments where multiple robots are operating simultaneously, VISOR can process several videos in parallel, limited only by the capacity of the VLM to handle multiple requests. Meanwhile, using VISOR frees human evaluators to focus on other tasks during this time, further improving overall efficiency. In contrast, a human can evaluate only one video at a time, making VISOR a more scalable solution for assessing thousands of tests. We note, however, that our current evaluation focused on videos generated from simulations. Consequently, the performance of VISOR on videos from hardware-in-the-loop simulations or real-world robotic deployments remains untested and represents an important direction for future work. At this stage, our focus is on simulation-based testing, which is the primary setting for large-scale evaluation of robotic applications. Accordingly, VISOR is designed to support automated test assessment at scale in simulation-based robotics CI/CD pipelines.

Finally, we anticipate that fine-tuning open-source VLMs on robotic task assessment data could improve their performance and reduce reliance on proprietary models, thereby reducing the monetary costs associated with proprietary models.

## 8 Related work

Recent research have enabled robots to perform complex manipulation tasks by integrating multimodal perception with action generation. However, evaluating these systems remains challenging in open-world settings, where failures stem from perceptual ambiguities, environmental variability, or underspecified instructions. Most existing approaches rely on benchmark-based evaluations with symbolic oracles [20, 27, 28, 36]. For instance, VLATest [50], which automatically generates scenes to evaluate VLAs, and large-scale benchmarks like VLABench [53] and Nebula [39], primarily report success rates focusing on final outcome correctness. However, these approaches offer limited insight into qualitative aspects such as motion quality, robustness, and perceptual grounding, while also relying on ad-hoc, task-specific test oracles. To address these limitations, Valle et al. [48] proposed quality and uncertainty metrics for

robotic task evaluation. In contrast to our approach, their method requires human intervention to set metric thresholds so that the metrics distinguish between different levels of execution quality.

A complementary research direction leverages Large Language Models (LLMs) and VLMs for failure detection and automated correction. REFLECT [30] converts multi-sensory robot actions into hierarchical textual summaries, enabling LLMs to explain failures and suggest corrective actions. RoboReflect [32] employs VLMs for reflective reasoning and trajectory planning adjustment in ambiguous grasping scenarios, while SC-VLA [24] integrates fast action prediction with a slower VLM-based system for detecting and correcting errors via chain-of-thought reasoning. Code-as-Monitor [54] leverages VLMs to generate code to monitor spatio-temporal constraints in both reactive and proactive modes of robotic tasks, and methods like RoboFAC [31] and AHA [15] fine-tune VLMs for natural-language reasoning over failures and trajectory corrections. Other approaches [14, 18] leverage VLMs for visual question answering, providing either binary task correctness or user-specific responses. While these methods demonstrate the utility of LLMs and VLMs for monitoring and recovery, they largely focus on failure correction or binary evaluation rather than assessment of execution quality.

In contrast to these approaches, VISOR evaluates not only task success but also execution quality, and provides uncertainty estimation to support trust in its verdicts. Moreover, unlike approaches that use symbolic oracles, VISOR can be easily adapted to other tasks, requiring no human intervention for threshold tuning or metric specification, but only for defining task quality requirements.

## 9 Conclusion and Future Work

Test oracle assessment for testing robotic behavior to determine its correctness remains a manual and, therefore, time-consuming task. To address this challenge, this paper proposes a VLM–based approach for automated test oracle assessment that determines not only whether a robot's behavior is correct but also its quality based on recorded robot behavioral videos. Moreover, VISOR explicitly quantifies uncertainty when assessing both task correctness and task quality, thereby providing a measure of how much the VLM's assessment can be trusted. We evaluate VISOR using two VLMs, GPT and Gemini, across four robotic tasks, analyzing their ability to assess correctness and quality while accounting for uncertainty.

As future work, we plan to evaluate World models like Cosmos from Nvidia [1]. These emerging models can support complex video analytics over large volumes of recorded and live video, enabling richer, more contextual understanding of visual content. Moreover, we will expand our evaluation to more VLMs and develop a voting-based mechanism to consolidate assessments from multiple VLMs.

## Acknowledgments

This work is supported by the InnoGuard Marie Skłodowska-Curie Doctoral Network (Grant Agreement No. 101169233). P. Arcaini is supported by the ASPIRE grant No. JPMJAP2301, JST. Pablo Valle and Aitor Arrieta are part of the Software and Systems Engineering research group of Mondragon Unibertsitatea (IT1519-22), supported by the Department of Education, Universities and Research of the Basque Country. Pablo Valle is supported by the Predoctoral Program for the Formation of Non-Doctoral Research Staff of the Education Department of the Basque Government (Grant n. PRE_2025_2_0252). We used generative AI to generate some code for the manuscript (e.g., LaTeX tables), implementation, and experiments.

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
