# OpenReview forum: "VISOR: A Vision-Language Model-based Test Oracle for Testing Robots"
_ACM.org/AIWare/2026/Conference — AIware 2026_

### Official Review · Reviewer_WZCS · 2026-03-09

**Rating:** 3
**Confidence:** 3

**Review:**

# Summary of strengths
> (1) The RQs are comprehensive and logically structured, covering effectiveness, uncertainty analysis, and the correlation between uncertainty and prediction quality.

> (2) The experimental setup includes multiple VLMs in the preliminary stage and compares two strong models in detail. The paper also adopts diverse evaluation metrics, including classification metrics, distance metrics, and uncertainty metrics.

> (3) The paper provides detailed statistical analyses, including repeated runs and statistical significance tests, which improve the reliability of the empirical findings.

# Summary of weaknesses
> (1) The proposed VISOR mainly relies on prompt engineering applied to existing VLMs without introducing substantial methodological innovations. The overall technical contribution appears relatively incremental.

> (2) All experiments are conducted on simulator-generated videos with a limited number of tasks and relatively simple scenarios. It remains unclear whether VISOR would perform effectively in more complex real-world robotic environments.

> (3) The evaluation mainly compares different VLMs rather than comparing VISOR with existing approaches for robotic test oracles. Even if existing methods only support binary task success evaluation, they could still serve as meaningful baselines.

# Detailed comments for authors
> (1) Quality: The paper is well structured and the empirical study is conducted rigorously. The dataset construction process and statistical analyses are carefully described. However, the methodological contribution itself is relatively limited, as the approach mainly relies on carefully designed prompts applied to existing VLMs.

> (2) Clarity: The paper is generally clear and easy to follow. The system overview and experimental setup are well explained.

> (3) Originality: The paper applies VLMs as test oracles for robotic testing, which is an interesting direction. However, the technical novelty is somewhat limited, since VISOR mainly consists of prompt design and evaluation of existing VLMs.

> (4) Significance: The problem of automated test oracles for robotic systems is important and challenging. The study provides useful empirical insights into the capabilities and limitations of VLMs for this task. However, the practical impact is somewhat limited by the simplified experimental setting, as all evaluations are performed on simulated environments.

# Questions for authors’ response:
> (1) How well does the paper expect VISOR to generalize to real-world robotic execution videos, where visual noise, lighting conditions, and environment complexity may differ significantly from simulation?

> (2) Could the paper include comparisons with existing baselines, even if they only support binary task success evaluation?

**Summary:**

This paper proposes VISOR, a VLM-based automated test oracle assessment approach for robotic system testing. The approach analyzes robot execution videos and determines whether a task is successfully completed and assesses its execution quality. VISOR also estimates the uncertainty of its predictions to provide additional reliability signals. The paper evaluates VISOR using two VLMs (GPT-4.1 and Gemini-2.5-Flash) on simulated robotic manipulation videos. The results show that Gemini achieves higher recall and GPT produces more precise predictions, while both models exhibit low uncertainty.

---

> ### Author Response · Authors · 2026-03-17
>
> We would like to thank the reviewer for a detailed and structured review. We would like to address the questions raised by the reviewer as follows:
>
> 1. While VISOR can be used for real-world videos and with more complex tasks, how well it performs remains an area of future investigation. However, our intention in this paper is to use simulation, the main technique for large-scale testing of robotic applications. Thus, VISOR is designed to support automated testing at scale (evaluating thousands of tests), primarily conducted in simulation within robotics CI/CD pipelines. Large-scale test executions (like the ones we execute) are not usually feasible in the real world. However, we believe this should be discussed in the paper, and we can easily add this explanation.
>
> 2. Using binary oracles as baseline would not be meaningful in our case because it relies on symbolic oracles, which we treat as ground truth in our evaluation. These symbolic oracles are assumed to be correct, as they are manually generated and used to label the data when assessing the effectiveness of our approach. Our motivation is that, if VLM-based approaches become sufficiently reliable, they could eventually replace symbolic oracles, removing the need to manually design them for each task and scene. This point can be easily added in the paper with minor editing.

---

### Official Review · Reviewer_VBcC · 2026-03-11

**Rating:** 2
**Confidence:** 3

**Review:**

# Pros

- The paper addresses the long-standing and important test oracle problem in robotic system testing. Reducing human effort in evaluating robotic task execution is a meaningful and well-motivated goal with clear practical value.
- The evaluation is conducted on a reasonably large and balanced dataset. The authors take care to control for class imbalance and report results across repeated runs.
- The use of appropriate statistical tests (e.g., Mann–Whitney U test, effect size analysis) strengthens the credibility of the empirical findings. Reporting both effectiveness metrics and uncertainty-related measures is a positive aspect of the work.

# Cons

- At its core, the proposed approach reduces to “using a VLM to watch execution videos and output task status and quality via prompting”. This framing closely follows the now-common LLM-as-a-Judge / VLM-as-a-Judge paradigm. While the application domain (robotic test oracles) is relevant, the paper does not introduce a substantially new oracle formulation, reasoning mechanism, or modeling insight beyond this paradigm.
- The experimental results clearly show the presence of both false positives and false negatives in task correctness classification. This naturally raises a fundamental question: if the oracle itself is unreliable, under what conditions is it usable for automated testing?

- The paper would benefit significantly from qualitative case studies that analyze why the VLM fails. For example: which failure modes are consistently missed by the VLM? Such analysis could provide actionable insights for improving VLM-as-a-Judge systems and would strengthen the paper’s contribution beyond reporting aggregate metrics.

**Summary:**

This paper proposes VISOR, a Vision-Language Model (VLM)–based approach to automated test oracle assessment for robotic systems. By prompting a VLM to judge execution videos, the approach aims to evaluate both task correctness (success vs. failure) and task quality (high/medium/low), while also reporting model uncertainty. The paper evaluates VISOR on a large, balanced dataset of simulated robotic manipulation videos using two proprietary VLMs, with extensive statistical analysis.

---

> ### Author Response · Authors · 2026-03-17
>
> We thank the reviewer for the feedback and address the identified cons point by point below.
>
> 1. We agree with the reviewer that the problem can be framed within the LLM-as-a-Judge/VLM-as-a-Judge paradigm. We would like to highlight that the existing oracles focus on evaluating task completeness. We, on the other hand, also judge the quality of the executed task. Thus, our oracle formulation is new. With minor edits in the paper, we can make it clearer.
>
> 2. We agree with the reviewer regarding the posed question. However, we would like to highlight that the focus of this study was to investigate whether existing models are suitable in our context and whether this test oracle is reliable. Naturally, our future work is to fine-tune them for this task to improve the test oracle's reliability. We can clarify this in the paper with minor editing.
>
> 3. As also suggested by other reviewers, we have now performed the suggested analysis and extracted case study examples.
>
>    - Regarding quantitative analysis, we found that the main missed failure types for the task correctness across the tasks are:
>         - Partial progress or near-completed tasks are often classified as success. We observed cases where the robotic arm ALMOST placed the object onto the target, but these borderline executions were still treated as successful.
>         - Robotic arm interacting with the wrong object is classified as success, like picking up the incorrect object. This may be attributed to the fact that sometimes it's hard to detect objects. For instance, in one such execution, we observed that the robotic arm picked up the Coca-Cola can and put it in the basket, although the task was meant to put the Pepsi can in the basket, and still the VLM classified it as successful.
>         - Robotic arm feigning the motion of the task is classified as success when the robotic arm is unable to complete one part of the task and proceeds to the next step.
>
>      These patterns are especially pronounced for Gemini, which misclassifies failure videos as success much more frequently than ChatGPT, particularly for MoveNear and PickUp tasks.
>
>    - Regarding the misjudgement of the quality of the execution videos by the VLM, below are the main factors behind these failures:
>        - Factors such as hesitation, vibration, unstable grasp, slight collision, or brief loss of control are often not reflected in the prediction. This causes the oracle to misjudge medium quality tasks to high quality, or low quality to medium or high quality. We observed cases like the robotic arm hesitated while moving the object near to the target object but still it was classified as High Quality by both the VLMs.
>        - VLMs sometimes ignore events that affect quality in the middle of the execution video and focus more on the end. For example, in one execution, while moving one object toward another, the orange rolled over upon contact with the arm near the target, but it was still classified as High Quality.
>        - VLM is sometimes unable to judge some borderline cases. In one such case during the “put in” task, an object was picked up by sliding only a little bit, but it was classified as medium quality instead of the intended high quality.
>        - Sometimes VLMs miss the issues in the end like dropping the object, hitting the object etc. When these issues happens near the end, the oracle may still predict a higher quality.

---

### Official Review · Reviewer_L8GQ · 2026-03-13

**Rating:** 3
**Confidence:** 3

**Review:**

Strengths:

+ The problem is well-motivated and practically relevant. Test oracle automation for robotic systems is an open challenge, and extending oracle assessment beyond binary pass/fail to quality grading is a meaningful contribution.

+ The experimental design is sound with 10 repeated runs, Mann-Whitney U tests with Vargha-Delaney effect sizes, balanced dataset construction, and a thoughtful custom distance metric that integrates correctness and quality into a single ordinal scale.

+ The open-source VLM exclusion is properly justified through a preliminary experiment.

+ The paper is clearly written and the replication package is provided.

Weaknesses.

- The most significant issue concerns the interpretation of the uncertainty analysis. RQ2 concludes that "both models exhibit consistently low uncertainty", presenting this as evidence of reliability. However, Section 5.2 acknowledges that the models remain highly confident even when incorrect, indicating an overconfidence problem. The abstract and conclusion highlight low uncertainty as a positive result without sufficiently qualifying this limitation.

- The results for task quality assessment (RQ1b) are generally weak. The best reported F1-micro score is 0.483 for GPT on the PutOn task, while most results fall below 0.40, and Gemini's distance metric frequently exceeds 2.0 out of 5. Although the study acknowledges this limitation, it does not investigate the causes. The quality-level decision rules are adopted directly from Valle et al. [48] and encoded as prompt strings. However, the study does not analyze whether these rules are unambiguous or whether VLMs interpret them consistently. A qualitative failure analysis mgiht improve the analytical depth of the study.

- The exclusion of all open-source VLMs is based on a preliminary experiment using only six videos per task, which seems insufficient to support a definitive exclusion. This is particularly relevant given the broader claims about the generality of the approach. Fine-tuned or instruction-tuned open-source models may behave differently, but this possibility is only briefly acknowledged in the discussion.

- The cost analysis is presented using informal reasoning rather than a structured comparison. The claim that VISOR provides scalability advantages over human evaluation is asserted but not empirically evaluatd.

**Summary:**

This paper proposes VISOR, which uses Vision–Language Models (VLMs), specifically GPT-4.1 and Gemini-2.5-Flash, as automated test oracles for robotic manipulation tasks. The system evaluates both task correctness (binary) and task quality (High/Medium/Low) based on simulation videos. The approach also quantifies VLM uncertainty. The evaluation includes 1,032 videos across four tasks derived from three Vision-Language Action (VLA) models.

---

> ### Author Response · Authors · 2026-03-17
>
> We thank the reviewer for the feedback and address the identified weaknesses point by point below.
>
> 1. This is a very good observation and can be easily addressed with minor edits to the paper.
>
> 2. We agree with the limitation mentioned. We have now investigated the causes, which are:
>      - VLMs often miss subtle events such as hesitation, unstable grasp, slight collision, brief loss of control, or small disturbances during execution.
>      - The three quality classes (High, Medium, Low) often differ only in fine-grained differences in execution quality from the video.
>      -  Some tasks are harder to interpret than others, like the move near task, where it's hard for the oracle to determine the spatial gap between objects.
>      -  VLMs sometimes ignore the events that affect the quality in the middle of the execution video.
>
> 3. We agree with the reviewer. However, we like to highlight that the focus of this study was to investigate existing models rather than fine-tuning them for this task, which is naturally our future work. In addition to fine-tuning, future work will evaluate more complex approaches, such as World Models and multi-agent architectures. However, it is important to note that the aim of this paper was to assess whether a simple and low-cost, in terms of computational resources, approach, could successfully perform this task.
>
> 4. We agree with the reviewer as well. However, as mentioned, such analysis is best performed empirically through human evaluation, as we need to measure the average human evaluation time to compare against the VLM evaluation time, which we aim to perform in the future.

---

> > ### Comment · Reviewer_L8GQ · 2026-03-18
> >
> > The authors have adequately addressed my comments and concerns.

---

### Official Review · Reviewer_UMhb · 2026-03-13

**Rating:** 2
**Confidence:** 3

**Review:**

Strengths:

- The oracle problem in robotic testing is underexplored.


Weakness:

- Automating quality assessment beyond binary correctness is a meaningful research direction. However, the contribution is essentially two prompt templates. The approach is zero-shot prompting of commercial APIs.

- Quality assessment results are underanalyzed. F1-micro for quality assessment ranges from 0.19 to 0.48 (Table 3). GPT on MoveNear achieves F1-micro of 0.193, which is below random guessing for a three-class problem. The paper acknowledges this briefly but offers no analysis of why the models fail. Quality assessment is the main differentiator from prior symbolic oracles. These results substantially weaken that claim.

- The paper provides no qualitative analysis or case study examples. A few illustrative cases would substantially strengthen the empirical findings. For example, showing which failure modes are consistently missed by the VLM or what distinguishes a low-quality execution from a medium one in practice.

- VLMs are known to be sensitive to domain shift. The gap between simulation rendering and real-world video could significantly affect results. For example, real-world robotic execution videos may exhibit properties absent in simulation, including motion blur, inconsistent lighting, cluttered backgrounds, sensor noise, and variable camera angles. These factors could substantially degrade VISOR's performance when applied outside a simulated environment. The paper acknowledges this as a threat but does not discuss it in the context of relevant sim-to-real literature.

**Summary:**

VISOR applies Vision-Language Models as test oracles for robotic manipulation. It assesses task correctness and quality from simulation videos. GPT and Gemini are evaluated on 1,032 videos across four tasks. The paper is clearly written. The problem is timely and relevant. However, the technical contribution is limited, and several core claims are not well supported.

---

> ### Author Response · Authors · 2026-03-17
>
> We thank the reviewer for the feedback and address the identified weaknesses point by point below.
> 1. We agree that the prompting component of our approach is centered on two prompt templates, one for task correctness assessment and one for task quality assessment. In addition, our contributions include uncertainty quantification, a comprehensive evaluation, including a study of the relationship between correctness and uncertainty, and a newly defined distance metric that penalizes the severity of test incorrectness. We can easily add these contributions explicitly in the introduction section. We also note that a simple approach to solving a complex problem (like this one) is more convenient than a complex one; our contribution goes beyond the technical approach, but also includes a large-scale empirical analysis. We believe this paper can serve as a starting point to target this problem.
>
> 2. The reviewer's point is good. The model fails mainly because of these reasons:
>    - VLM capability is limited to reasoning about the spatial changes.
>    - The three quality classes (High, Medium, Low) often differ only in fine-grained differences in execution quality from the video.
>    - Some tasks are harder to interpret than others, like the move near task, where it's hard for the oracle to determine the spatial gap between the objects.
>
> This analysis can be easily added to the paper. Please note that, in addition to typical metrics, we also introduced a distance metric to address a three-class problem and to measure how incorrect they are relative to the ground truth.
>
> 3. This is a very good suggestion. We have now performed the suggested analysis and extracted case study examples.
>
>    - Regarding quantitative analysis, we found that the main missed failure types for the task correctness across the tasks are:
>         - Partial progress or near-completed tasks are often classified as success. We observed cases where the robotic arm ALMOST placed the object onto the target, but these borderline executions were still treated as successful.
>         - Robotic arm interacting with the wrong object is classified as success, like picking up the incorrect object. This may be attributed to the fact that sometimes it's hard to detect objects. For instance, in one such execution, we observed that the robotic arm picked up the Coca-Cola can and put it in the basket, although the task was meant to put the Pepsi can in the basket, and still the VLM classified it as successful.
>         - Robotic arm feigning the motion of the task is classified as success when the robotic arm is unable to complete one part of the task and proceeds to the next step.
>
>      These patterns are especially pronounced for Gemini, which misclassifies failure videos as success much more frequently than ChatGPT, particularly for MoveNear and PickUp tasks.
>
>    - Regarding the misjudgement of the quality of the execution videos by the VLM, below are the main factors behind these failures:
>        - Factors such as hesitation, vibration, unstable grasp, slight collision, or brief loss of control are often not reflected in the prediction. This causes the oracle to misjudge medium quality tasks to high quality, or low quality to medium or high quality. We observed cases like the robotic arm hesitated while moving the object near to the target object but still it was classified as High Quality by both the VLMs.
>        - VLMs sometimes ignore events that affect quality in the middle of the execution video and focus more on the end. For example, in one execution, while moving one object toward another, the orange rolled over upon contact with the arm near the target, but it was still classified as High Quality.
>        - VLM is sometimes unable to judge some borderline cases. In one such case during the “put in” task, an object was picked up by sliding only a little bit, but it was classified as medium quality instead of the intended high quality.
>        - Sometimes VLMs miss the issues in the end like dropping the object, hitting the object etc. When these issues happens near the end, the oracle may still predict a higher quality.
>
>
> 4. While VISOR can be used for real-world videos, how well it performs remains an area of future investigation. However, our intention in this paper is to use simulation, the main technique for large-scale testing of robotic applications. Thus, VISOR is designed to support automated testing at scale (evaluating thousands of tests), primarily conducted in simulation within robotics CI/CD pipelines. However, we believe this should be discussed in the paper, and we can easily add this explanation.